# Effects of Early Rehydration on Brain Perfusion and Infarct Core after Middle Cerebral Artery Occlusion in Rats

**DOI:** 10.3390/brainsci11040439

**Published:** 2021-03-29

**Authors:** Yuan-Hsiung Tsai, Chia-Hao Su, I-Neng Lee, Jen-Tsung Yang, Leng-Chieh Lin, Yen-Chu Huang, Jenq-Lin Yang

**Affiliations:** 1Department of Diagnostic Radiology, Chiayi Chang Gung Memorial Hospital, Chiayi 613016, Taiwan; russell.tsai@gmail.com; 2Institute for Translation Research in Biomedicine, Kaohsiung Chang Gung Memorial Hospital, 123 Ta Pei Road, Kaohsiung City 83301, Taiwan; chiralsu@cgmh.org.tw; 3Medical Research, Chiayi Chang Gung Memorial Hospital, Chiayi 613016, Taiwan; geneasylum@yahoo.com.tw; 4Department of Neurosurgery, Chiayi Chang Gung Memorial Hospital, Chiayi 613016, Taiwan; jents716@ms32.hinet.net; 5Department of Emergency Medicine, Chiayi Chang Gung Memorial Hospital, Chiayi 613016, Taiwan; a3456711@ms65.hinet.net; 6Department of Neurology, Chiayi Chang Gung Memorial Hospital, Chiayi 613016, Taiwan; yenchu.huang@msa.hinet.net

**Keywords:** ischemic stroke, perfusion, dehydration, hydration

## Abstract

Imaging evidence for the effect of rehydration on cerebral perfusion and brain ischemia has never been proposed in the literature. This study aimed to test the hypothesis that early rehydration treatment can improve cerebral perfusion and decrease infarct volume, consequently reducing mortality of dehydrated stroke animals. Methods: Thirty dehydrated experimental rats were randomly assigned to either a rehydration or control group after middle cerebral artery occlusion (MCAO). Diffusion-weighted imaging and dynamic contrast enhancement perfusion imaging were performed at 30 min and 6 h after MCAO using a 9.4T MR imaging scanner to measure the infarct volume and brain perfusion. Results: The survival rates after the first MRI scan were 91.7% for the rehydration group and 58.3% for the control group (*p* = 0.059). The survival rates after the second MRI scan were 66.7% for the rehydration group, and 8.3% of the control group survived (*p* = 0.003). The infarct volume of the rehydration group was significantly smaller than control group at 30 min after MCAO (*p* = 0.007). The delay time and time to maximum were significantly shorter in the rehydration group at 30 min (*p* = 0.004 and 0.035, respectively). Conclusions: The findings suggest that early rehydration therapy can decrease the infarct volume, shorten the delay time of cerebral perfusion, and increase survival of dehydrated ischemic-stroke rats. This preliminary study provided imaging evidence that more intensive early hydration therapies and reperfusion strategies may be necessary for acute stroke patients with dehydrated status.

## 1. Introduction

Dehydration has a high prevalence in ischemic stroke and is associated with stroke-in-evolution and functional outcome [1,2,3,4]. The reasons for these observations are based on the concept that dehydrated status may reduce brain perfusion, decrease collateral blood flow, and aggravate ischemic brain injury [5,6,7,8]. Current acute stroke guidelines recommend rehydration for patients who are under dehydrated situations [9]. Previous studies suggested rehydration could improve the outcome and reduce death rates in dehydrated acute stroke [10,11]. However, the imaging evidence for the effect of rehydration on cerebral perfusion and brain ischemia has never been discussed in the literature.

Diffusion-weighted imaging (DWI) identifies tissue wherein diffusion is restricted as a result of cytotoxic edema, which leads to a reduction in the apparent diffusion coefficient (ADC). DWI is, by far, the most sensitive and accurate imaging method used to detect acute ischemic stroke. The restricted diffusion in ischemic brain tissue can be observed within a few minutes after arterial occlusion [12]. Dynamic susceptibility contrast (DSC) perfusion-weighted imaging measures brain perfusion and has been regarded as a reliable tool for evaluating hyperperfused and penumbra areas in many stroke trials [13].

This study aimed to investigate the effect of rehydration therapy in the cerebral perfusion and infarct core using an acute middle cerebral artery occlusion (MCAO) rat model. DWI and DSC imaging were used to measure the size of the infarct core and cerebral blood perfusion, respectively. We hypothesized that rehydration therapy at the acute stage of ischemic stroke can improve brain perfusion, reduce further ischemic brain injury, and increase the survival rate.

## 2. Materials and Methods

The flow diagram of the study design is illustrated in Figure 1. Thirty male eight-week-old Sprague-Dawley rats were obtained from BioLASCO Taiwan Co., Ltd. (Taipei, Taiwan); they were maintained at a constant day/night (12/12 h) period, temperature (21 ± 2 °C), and humidity (50–60%) with free access to food and water in an animal facility recognized by the Association for Assessment and Accreditation of Laboratory Animal Care International (AAALAC International). The experimental procedures were approved and supervised by the Animal Committee of Chang Gung Memorial Hospital (Approval No. 2017082101) and followed the guidelines for animal care and use.

### 2.1. Dehydration and Ischemic Stroke Rat Model

For preparation of dehydration animals, rats were housed in a 2100-R rodent metabolic cage (Lab Products, Inc., Seaford, DE, USA) providing sufficient food but without water for 24 h prior to the MCAO surgery. The urine of dehydration animals was collected during the 24 h dehydration period for lab testing to determine whether they achieved dehydrated status [14,15]. Focal cerebral ischemia was induced via MCAO procedure, which followed the previously described method of Chen et al. [16]. Rats were anesthetized by intraperitoneal injection with sodium pentobarbital (50 mg/kg body weight (bw)). The MCAO procedure is briefly described as following. The right eye-to-ear area was shaved. The rats were then placed in a prone position on a warming pad at 37 °C and incubated with positive-pressure ventilation (0.2 mL/sec) with oxygen using a small animal ventilator (SAR-830/A; CWE, Ardmore, PA, USA). An incision of around 1.5 cm was made on the scalps at the midpoint between the right eye and right ear. The temporalis muscle was then separated to expose the zygoma and squamosal bones. After that, a dental drill was used to make a 2 mm^2^ burr hole that was above 1 mm rostral to the anterior junction of the zygoma and the squamosal bones. The dura mater was carefully pierced with a microsurgical needle to expose the middle cerebral artery (MCA) which was carefully isolated and ligated using 10.0 surgical sutures (Johnson & Johnson Medical, Somerville, NJ, USA) to induce ischemic stroke in the right hemisphere. Isoflurane (2%) was on hand in case the rats woke up during surgery. Once the MCA ligation was complete, the bilateral common carotid arteries (CCAs) were immediately ligated by aneurysm clips. The ligations on both the CCAs and the MCA were loosened after 30 min. All procedures were completed in 90 min.

Rats were randomly assigned to either the rehydration or control (without immediate hydration) group. Each rat in the rehydration group received intraperitoneal infusion of 1.5 mL saline each at 3 time points: immediately after the MCAO surgery, after the first MRI scan, and before the second MRI. All experimental rats were sent for the first MRI scan approximately 30 min after surgery. The rats were anesthetized again 6 h after surgery for a second set of MRI scans. Between the first and the second MRI scans, rats had free access to water. The rats were sacrificed with overdose of isoflurane after the second MRI scan.

### 2.2. MRI Acquisition and Analysis

MRI was acquired by a 9.4 Tesla horizontal-bore animal MR scanning system (Biospec 94/20, Bruker, Ettlingen, Germany). This scanner comprises a self-shielded magnet with a 20 cm clear bore and a BGA-12S gradient insert of 12 cm inner diameter that offers a maximal gradient strength of 675 mT/m with a transmitter-only coil and a receiver rat brain surface array coil for signal detection. The MRI images were acquired at 30 min and 6 h post MCAO for each rat to generate the DWI ADC and DSC maps. Before the MR scans, animals were anesthetized with 3.0% isoflurane in room air and were placed inside the magnet. The anesthesia delivery to the rats was then maintained using a gas mixture of ~1.5–2.0% isoflurane and oxygen during the MRI scans. Animals were physiologically monitored throughout the MR imaging experiments.

DWI scans were obtained to produce the DWI and ADC maps using spin-echo EPI with the following parameters: TR = 3000 milliseconds (ms); TE = 27 ms; matrix size = 128 × 128; field of view = 2 × 2 cm^2^; number of slices = 9, which were 1 mm thick with a 0.5 mm gap; three directions = x, y, z; and b values = 0, 100, 300, 500, 700, 800, and 1000 s/mm^2^. DSC were acquired using a gradient echo EPI with the following parameters: TR = 25 ms; TE = 4 ms; matrix size = 96 × 96; field of view = 2 × 2 cm^2^; flip angle = 20 degree band width; 250 kHz; and 65 measurements, with an intravenous bolus injection of gadolinium contrast agent (0.5 mmole/kg) at the 6th dynamic. The total MRI scan time was 13 min and 48 s, which included a DWI scan of 11 min 12 s and a DSC scan of 2 min 36 s.

The image data were processed by commercial software MIStar software (Apollo Medical Imaging Technology, Melbourne, VIC, Australia). Cerebral blood volume, cerebral blood flow, mean transit time (MTT), delay time and time to maximum (Tmax) maps were generated using DSC-MRI mode with stroke-stenosis model in MIStar (Figure 2 and Figure 3). The relative cerebral blood flow (rCBF) and relative cerebral blood volume (rCBV) were calculated based on the automatically derived normal values. The indices of brain perfusion were measured by manual drawing region of interest (ROI) of the cortex of infarcted hemisphere on the rCBV image using the MIStar ROI tool. A mirror ROI on the contralateral healthy hemisphere was then generated automatically by the software. These two ROIs were applied to other perfusion maps to measure other perfusion indices. The volume of the infarction was manually measured on the DWI map using polygon selection.

### 2.3. Statistical Analysis

The Statistical Program for Social Sciences (SPSS) statistical software (version 18, Chicago, IL, USA) was used for the statistical analyses. Continuous variables were expressed as the mean ± standard deviation and were compared by performing Mann–Whitney U tests. Categorical variables were compared using the chi-square test or Fisher’s exact test. The level of statistical significance was set at *p* < 0.05.

## 3. Results

Six rats died during or after the MCAO surgery, and the surviving 24 rats were randomized into rehydration and control groups. The results are shown in Table 1. The urine indices of both group were all high to the dehydrated levels [14,15]. The urine osmolality and specific gravity were significantly higher in the rehydration group (*p* = 0.016 and 0.025, respectively), suggesting a more severely dehydrated status of the rehydrated group (Table 1). A rat of the rehydration group died during the first MRI scan, and three died before the second MRI scan. Five rats in the control group died during the first MRI scan, five died before and one died during the second MRI scan. Thus, the survival rates at the time after the first MRI were 91.7% for the rehydration group and 58.3% for the control group (*p* = 0.059). The survival rates at the time after the second MRI were 66.7% for the rehydration group and only 8.3% for the control group (*p* = 0.003) (Figure 4).

The volume of the infarction defined by the DWI map at 30 min post MCAO was significantly smaller in the rehydration group than in the control group (*p* = 0.007). The delay time and Tmax at 30 min were significantly shorter in the rehydration group than in the control group (*p* = 0.004 and 0.035, respectively). However, other perfusion indices, including rCBV, rCBF, and MTT at 30 min, as well as the volume of infarction and all perfusion indices at 6 h, were not significantly different between the two groups (Figure 4).

## 4. Discussion

In the animal study, we tested the hypothesis that early rehydration therapy would improve brain perfusion, reduce infarct volume, and thus increase survival. Acute stroke guidelines suggest the use of rapid repletion of intravascular volume for ischemic stroke patients, especially for those who are hypovolemic at presentation [9]. There have been only a few studies of rehydration for acute ischemic stroke patients with dehydrated status. Small series studies by Lin et al. showed that rehydration of dehydrated stroke patients with isotonic saline could lower rates of stroke in evolution and infection and shorten lengths of hospital stay [17,18]. However, no human neuroimaging study has been proposed to provide direct evidence of the benefit of rehydration to brain perfusion and ischemic brain tissue. This is due to the limitations often encountered in human stroke studies, in which it is not easy to control all the associated factors, such as the time of imaging after stroke onset, stroke subtype, accurate state of hydration at the time of stoke, and hydration therapy before the imaging studies were completed. These are even more difficult in cases of intra-venous thrombolysis or intra-arterial thrombectomy for acute stroke management. In the current study, we evaluated the volume of infarction and cerebral perfusion with a dehydration and MCAO-induced ischemic rat model using high field MRI to better control for other factors that affect cerebral perfusion and infarct volume. The results demonstrate that early rehydration could reduce the high mortality rate of dehydrated ischemic rats, decrease infarct volume, and improve delay time and Tmax in the ischemic hemisphere. By adding a imaging evidence, this finding extends previously reported work indicating that hydration therapy is important in acute stroke management for patients with dehydration [8,9,19,20].

Dehydration at the time of stroke has been acknowledged as being associated with early neurological deterioration and poor functional outcome [2,3]. In our previous animal study with a dehydration protocol of water deprivation for 48 h, approximately 50% of the rats died after MCAO surgery [6]. In the current study, we shifted the dehydration protocol to water deprivation for 24 h and confirmed the status of dehydration with urine analysis, but the mortality rate immediately after MCAO surgery was maintained high (20%) and even slightly higher in the following stages of study. Only two rats without early rehydration after survived 6 h, and the survival rate was improved via rehydration therapy. Another animal study supported the finding that access to water and food was independently associated with decreased mortality in mice with induced infarction [21]. Dehydration is known to cause increased blood viscosity, reductions in cardiac stroke volumes, and decreased blood pressure [22,23]. It may also lower the blood flow of the kidney, muscle, and carotid artery and reduce cardiac output [24,25,26]. All these factors lead to a higher mortality of dehydrated rats, especially after suffering severe brain ischemic injury and prolonged general anesthesia. Although extreme dehydrated status, such as in the animal model, is seldom encountered in human subjects, given the high prevalence of dehydration in stroke patients, care givers should be aware of patients’ state of hydration and that early rehydration is important for acute stroke management.

In this study, there were significant reductions in Tmax and delay time in the rehydration group. Tmax represents the time to maximum of the residue function obtained by deconvolution and has been used in recent clinical trials as the most appropriate perfusion parameter to represent the penumbra. Delay time is a specific perfusion parameter generated by the MIStar using the delay- and dispersion-corrected singular value deconvolution (ddSVD) method. In comparison with Tmax, delay time has correction of both delay and dispersion, which may be more precise to reflect the pathophysiological process of the time for contrast to travel from the selected arterial input function ROI to the tissue voxel [27,28]. Dehydrated status increases blood viscosity and decreases cerebral blood flow [29]; it has also been shown to impair cerebral autoregulation such as it has led to greater changes in mean middle cerebral artery velocity compared with normal hydration following exercise and during a cold pressure test [30,31]. Dehydration has also been shown to reduce the development of leptomeningeal collaterals after occlusion of the middle cerebral artery in stroke patients [7]. Furthermore, the prolonged delay time or Tmax may be the consequence of tissue damage and edema, which delay the blood inflow to the tissue. Other perfusion indices, including rCBF, rCBV, and MTT did not result in statistically significant differences between rehydration and control groups in this study. These three physiologic perfusion indices changed differently after occlusion of the artery in brain regions suffered from different levels of ischemic injury (such as the ischemic core, penumbra, and benign oligemic regions) [32]. Studies with a dynamic measurement of brain perfusion in each brain region may be more accurate in reflecting the effect of rehydration. Furthermore, further studies to understand if rehydration therapy can improve cardiac output, fluid volume, and collateral flows, reducing ischemic brain injury, are necessary.

Nevertheless, our study had several limitations. First, it took almost 14 min to complete the DWI and DSC scans. As we know that ischemic stroke is a dynamic process, the damage of brain tissue progresses every second after the occlusion of the artery. The infarct volume and tissue perfusion should be changing during scanning. It usually takes less than 5 min to obtain DWI and DSC images for stroke patients with a human MRI scanner, but the procedure needs more time for animals when using a high-field MRI. In order to shorten the scan time, we did not include other conventional MRI sequences into the scanning protocol. The second limitation is the relatively small sample size, which is also due to the high mortality rate of the MCAO rats. The high mortality rate also produces a potential study bias because rats with larger infarctions or more severe decreases in cerebral perfusion might be excluded. Third, the rats were randomized into two groups after MCAO surgery, but the results of the urine analysis showed a more severely dehydrated status in the rehydration group (Table 1). However, this imbalance might not change our results because the rats in the rehydration group had better brain perfusion and smaller infarction, even though they were more severely dehydrated before surgery. However, another control group with normal hydration and without water deprivation conduces a better understating of how early rehydration therapy could improve brain perfusion and protect the brain from ischemic injury.

## 5. Conclusions

In conclusion, our results demonstrate that early rehydration therapy for MCAO rats with dehydrated status can decrease the infarct volume and improve survival possibilities via the improvement of brain perfusion. This study provided an imaging clue that more intensive hydration therapies and aggressive reperfusion strategies might be necessary for the acute management of ischemic stroke patients with dehydrated status.

## Figures and Tables

**Figure 1 brainsci-11-00439-f001:**
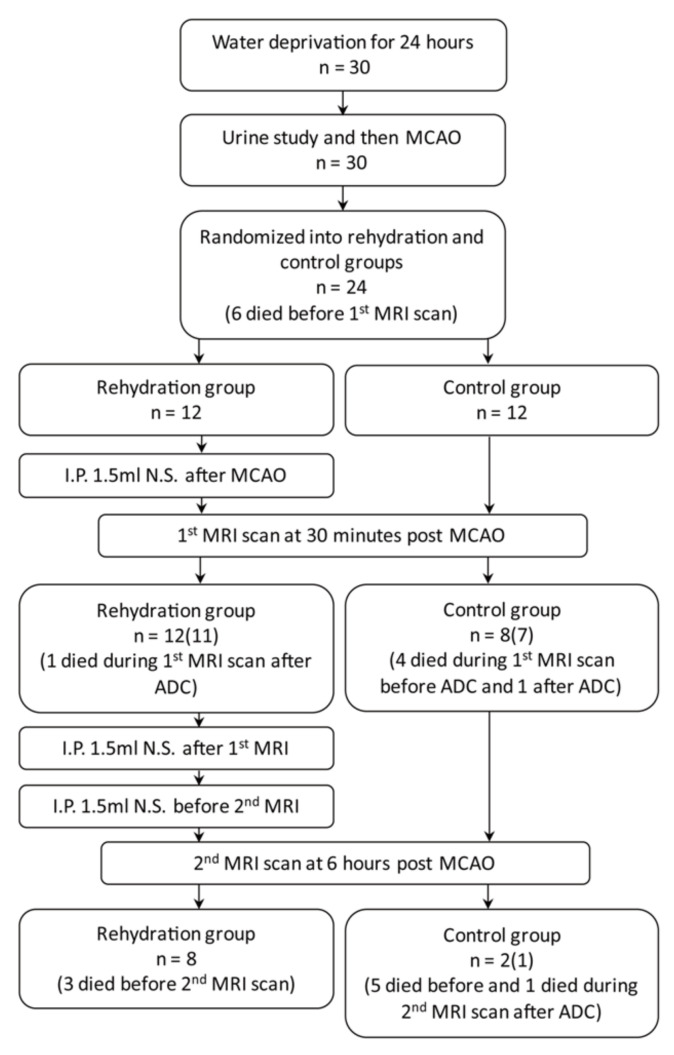
A flowchart of study workflow. (Case numbers in the parenthesis represent the number of rats completed the MRI scan with perfusion imaging). MCAO, middle cerebral artery occlusion; ADC, apparent diffusion coefficient.

**Figure 2 brainsci-11-00439-f002:**
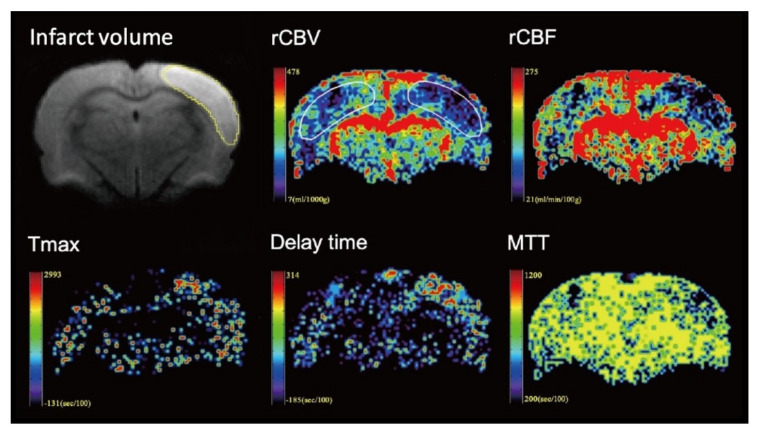
Representative images from one rat of the control group. The volume of the infarction was manually measured on the diffusion-weighted imaging (DWI) map. To measure the perfusion indices, the region of interest (ROI) was manual drawing in the infarcted hemisphere on the relative cerebral blood volume (rCBV) image. A mirror ROI on the contralateral healthy hemisphere was then generated automatically by the software. The ROIs were applied to other perfusion maps to measure other perfusion indices.

**Figure 3 brainsci-11-00439-f003:**
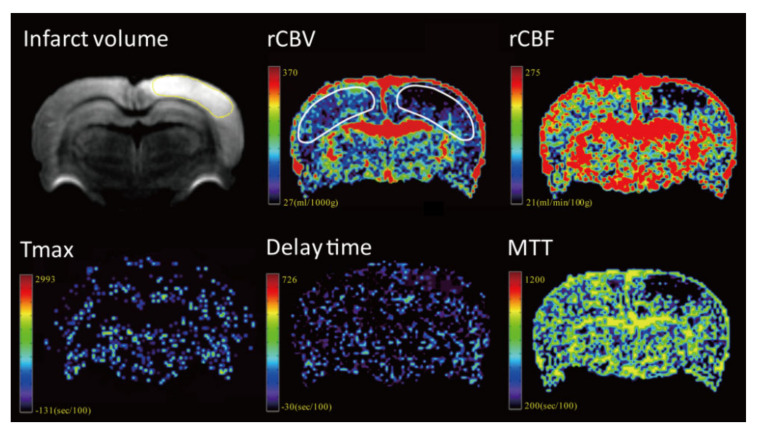
Representative images from one rat of the rehydration group.

**Figure 4 brainsci-11-00439-f004:**
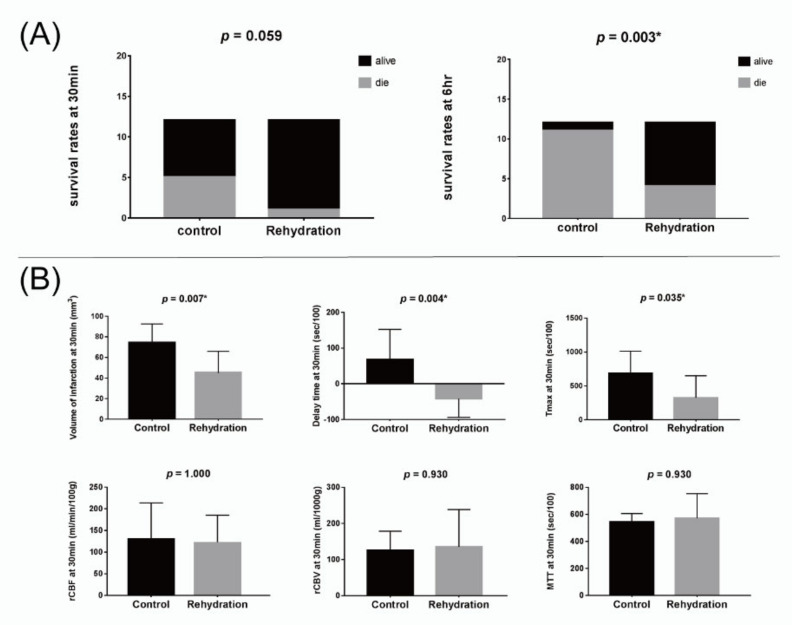
(**A**) Bar charts for the survival rates of the control and rehydration groups at 30 min (left) and 6 h (right) after surgery. (**B**) Bar charts for the perfusing indices of the control and rehydration groups at 30 min after surgery. * *p* < 0.05.

**Table 1 brainsci-11-00439-t001:** Comparison of the infarct volume and brain perfusion in the two groups.

	ALL		Control Group		Rehydration Group		*p-*Value
	Mean	SD	Mean	SD	Mean	SD	
Urine sodium (mg/dL)	303.35	58.91	308.13	44.01	300.17	68.81	0.678
Urine osmolarity (mosm/KgH2O)	2776.15	513.24	2493.75	291.47	2964.42	551.60	0.016 *
Urine specific gravity	1.07	0.02	1.07	0.01	1.07	0.02	0.025 *
MRI results at 30 min post MCAO							
Volume of infarction (mL)	56.37	24.48	73.72	18.73	44.80	21.15	0.007 *
Infarct side							
rCBF (mL/min/100 g)	126.51	68.25	131.82	81.84	123.14	62.19	1.000
rCBV (mL/1000 g)	131.62	85.37	125.52	53.12	135.50	103.21	0.930
Mean transit time (MTT) (sec/100)	560.61	145.11	543.50	63.05	571.49	181.88	0.930
Delay time (sec/100)	0.99	84.60	68.39	84.28	−41.90	51.98	0.004 *
Tmax (sec/100)	464.95	365.18	685.72	329.14	324.46	325.39	0.035 *
Normal side							
rCBF (mL/min/100 g)	190.57	141.81	244.69	173.47	156.13	113.04	0.211
rCBV (mL/1000 g)	192.36	137.98	211.10	94.10	180.44	163.25	0.151
MTT (sec/100)	645.80	144.83	602.72	178.69	673.22	119.90	0.479
Delay time (sec/100)	−59.56	88.72	−21.55	117.01	−83.74	59.25	0.425
Tmax (sec/100)	226.89	152.15	279.86	177.59	193.18	131.21	0.211
MRI results at 6 h post MCAO							
Volume of infarction (mm^3^)	82.83	48.83	147.12	35.74	66.76	37.52	0.089
Infarct side							
rCBF (mL/min/100 g)	117.66	92.29	40.78		127.26	93.72	0.444
rCBV (mL/1000 g)	129.92	117.08	36.12		141.65	119.38	0.444
MTT (sec/100)	515.75	117.84	392.44		531.16	115.87	0.444
Delay time (sec/100)	6.80	47.99	−1.82		7.88	51.19	1.000
Tmax (sec/100)	500.85	318.88	295.21		526.55	330.77	0.444
Normal side							
rCBF (mL/min/100 g)	139.78	47.13	116.11		142.74	49.48	0.667
rCBV (ML/1000 g)	148.81	61.39	92.61		155.83	61.64	0.444
MTT (sec/100)	598.91	110.14	540.36		606.23	115.38	0.444
Delay time (sec/100)	−41.03	64.50	−95.24		−34.25	65.44	0.667
Tmax (sec/100)	341.72	168.36	130.54		368.12	158.83	0.222

* *p* < 0.05.

## Data Availability

Data are contained within the article. The data presented in this study are available in the Table 1.

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
