# Peer review of "Effects of Early Rehydration on Brain Perfusion and Infarct Core after Middle Cerebral Artery Occlusion in Rats"

_brainsci, 2021, doi:10.3390/brainsci11040439_

Round 1

Reviewer 1 Report

The study purpose is to study the effects of early rehydration on brain perfusion and infarct core after MCAO surgery in rats. Publication may be recommended after some additional information, which makes it more convincing that the method would actually work with reproducible results. Detailed comments, suggestions and questions are given below.

  1. Why did you take MRI imaging after 30 min. and 6 hours after MCAO? Are there any special reasons?
  2. In sections 2-1, you explained about MCAO procedure. As I know, there is another MCAO method that accesses MCA through the neck. Are there any special reasons that you used the described MCAO method?
    3. In figure 2, the result MRI and TTC staining images are shown. Are those images from the rehydration group? Please, show and compare the results images from both control and rehydration cases.
    4. How long does it take to get one MRI image? I think that it takes a long time to get all MRI images of all samples. Then, there is probably a significant time difference between the first sample and the last sample. Any issues on this?
  3. In the 253rd line, TTC staining is used to confirm the final infarct volume. Was there an infarct volume difference between the groups at each stage, for example after 1st MRI and after 2nd MRI? It can be better to compare the TTC results with the MRI results?

Please, check these typos

  1. Page 7; In Fig.3, make the axis labels bigger.

Author Response

  1. Why did you take MRI imaging after 30 min. and 6 hours after MCAO? Are there any special reasons?

Response: The reasons we scanned the rats at 30 mins and 6 hours after MCAO include:

  1. For the first time point, we wanted to see the effect of dehydration on brain tissue immediately after occlusion of the artery. As we know that time is the brain, time loss is brain loss for acute stroke patients, thus identifying the physiological change of the brain during hyper-acute stage is important. However, we needed time to prepare the rats ready for MRI scan after MCAO and 30 mins was the earliest time to be achieved.
  2. There are two reasons we selected 6 hours as the 2nd time point. First, in recent large series clinical trials for acute stroke therapy, (mostly for intra-arterial thrombectomy, such as the MR CLEAN, EXTEND-IA, ESCAPE, SWRIT PRIME trials and others) (for review article, please see Lancet. 2016 Apr 23;387(10029):1723-31.) emergent reperfusion therapies performed within 6 hours after onset were considered to be effective and with less complications. Second, we did MRI scan 4 hours after MCAO in our previous study (Front Neurol. 2018 Sep 20;9:786.) and felt that some rats did not recover well from general anesthesia after the 1st MRI scan and before the 2nd scan. Thus, it was not easy to monitor the behavior and the mortality rate might be high due to repeated anesthesia in a short period of time.
  3. In sections 2-1, you explained about MCAO procedure. As I know, there is another MCAO method that accesses MCA through the neck. Are there any special reasons that you used the described MCAO method?

Response: The method reviewer has mentioned may be the one described by Longa et al. (Stroke (1989) 20:84–91.). After the common carotid artery, the external carotid artery, and the internal carotid artery are sequentially exposed, we can deliver a polyamide

monofilament non-absorbable surgical suture via the ECA into the ICA until the bifurcation of the middle cerebral artery and the anterior cerebral artery to block the circulation. Blood flow can be restored (reperfusion) by the withdrawal of the inserted suture after 1 hour of vessel occlusion to mimic the establishment of collateral circulation or reperfusion in acute stroke patients. We did this kind of MCAO model before (also presented in Front Neurol. 2018 Sep 20;9:786.) but found the area or volume of infarction were more heterogeneous among MCAO rats thus the effect of the controlled variables ( such as Rehydration in the current study) might not be significant in a study with small sample size. Thus, we shifted to the current MCAO model.  

  1. In figure 2, the result MRI and TTC staining images are shown. Are those images from the rehydration group? Please, show and compare the results images from both control and rehydration cases.

Response: The images presented in figure 2 were from one rat of the control group. We showed these images to demonstrated how we measured the infarct size and perfusion indices. Following this comment, we have added another figure (figure 3) representing images from one rat of the rehydration group into the revised manuscript.

  1. How long does it take to get one MRI image? I think that it takes a long time to get all MRI images of all samples. Then, there is probably a significant time difference between the first sample and the last sample. Any issues on this?

Response: The total MRI scan time was 13 minutes and 48 seconds which include a DWI scan of 11 minutes 12 seconds and a DSC scan of 2 minutes 36 seconds following contrast injection. We thank the reviewer for the careful review and for pointing out this important issue. As ischemic stroke is a dynamic process, the damage of brain tissue changes every second after occlusion of the artery. It usually takes less than 5 minutes to obtain DWI and DSC images for a stroke patient with a human MRI scanner but needs more time for animals with a high-field MRI. In order to shorten the scan time, we did not include other MRI sequences, such as T1WI and T2WI images into the scan protocol. We have added a short discussion about this into the limitation section and the scan times for DWI and DSC have been added into the methods section (2-2 MRI acquisition and analysis).

  1. In the 253rd line, TTC staining is used to confirm the final infarct volume. Was there an infarct volume difference between the groups at each stage, for example after 1st MRI and after 2nd MRI? It can be better to compare the TTC results with the MRI results?

Response: We thank the reviewer for this important comment. Our initial study design was to use TTC staining as a standard for final infarct volume and also planned to compare the TTC with MRI/ DWI result. However, as we mentioned in the limitation section, the time for TTC study was diverse for each rat; thus, the results were not suitable for group comparison. Some of the rats, especially in the control group, died before the study was completed at 6 hours after MCAO. Follow the recommendation of another reviewer, the presentations about TTC, including those in the methods, results, discussion sections as well as in the figure and table, have been removed in the revised manuscript.

Please, check these typos

  1. Page 7; In Fig.3, make the axis labels bigger.

Response: The font size of the axis labels has been enlarged.

Reviewer 2 Report

Dehydration is a prominent part of ischemic stroke pathology. American Heart/Stroke Association noticed it long back and recommended rehydrating the dehydrated patients (Ref 9-11, this manuscript). Similar to observations in stroke patients, dehydration has been reported in animal models of stroke.

            Using a permanent ischemic stroke rat model, this manuscript reports the harmful effects of dehydration. The study provides MRI-based evidence that rehydration is beneficial. It protects the ischemic brain against infarct injury and enhances the rate of survival. MRI studies within a short period (30 min-6h) after human stroke is less feasible.  Therefore, animal MRI studies showing the efficacy of rehydration are significant.

            One can understand the difficulty of performing MRI procedures twice within six h. As shown, the death rate is also high, and ultimately, only one animal is left in the control group. This limitation renders the study weak. TTC studies are also not acceptable based on one animal. I will suggest removing the TTC staining as it does not add much. Overall, the quality of MRI studies is good. Despite several weaknesses, I believe that having published MRI data on the efficacy of rehydration in ischemic stroke may help the stroke clinic.

Author Response

Response: We thank the reviewer for the careful review and understand the reviewer’s concerns regarding the TTC staining. Our initial study design was to use TTC as a standard for final infarct volume and would also compare the TTC staining with MRI/ DWI result. However, as we mentioned in the limitation section, the time for TTC study was diverse for each rat. Some of the rats, especially in the control group, died before the study was completed at 6 hours after MCAO. Thus, the results were not adequate for further analysis.  Follow your recommendation, the presentations about TTC staining, including those in the methods, results, discussion sections as well as in the figure and table, have been removed in the revised manuscript.